# Selectivity and Market Timing Ability of Fund Managers: Comparative Analysis of Islamic and Conventional HSBC Saudi Mutual Funds

**Marwa Zouaoui** [1,2]

1   Department of Finance, Northern Border University, Arar 91431, Saudi Arabia; marwazoua@gmail.com;
    Tel.: +966-055-7717-052
2   Department of Finance, ISG Tunisia, University of Tunis, Tunis 20000, Tunisia

**Abstract:** This paper empirically compares the market timing, the stock selection and the performance persistence of Islamic and conventional HSBC Saudi mutual funds by using monthly returns from April 2011 to December 2018. The data was grouped into five portfolios based on geographical investment basis (locally, Arab, internationally) and Sharia compliance (Islamic and conventional). The empirical results indicate that Islamic funds underperformed conventional funds internationally but not locally. Findings suggest that the market selectivity skills of managers in the Islamic funds are better than the conventional funds. In addition, only the managers of Saudi conventional funds investing internationally have a good market timing skills, thus, they are able to beat the market index by predicting its movements and buying and selling accordingly. Furthermore, this study gives a brief idea about the performance persistence of HSBC Saudi funds. The results confirm existence of the persistence performance when the funds do not apply Sharia law and when they are instead focused internationally.

**Keywords:** Islamic mutual funds; performance analysis; performance persistence; market timing; selectivity skills; GARCH modeling

## 1. Introduction

Analyzing the performance is extremely important to understand investment funds operation, but knowing if this performance could persist in the future is even more attracting. The study of performance persistence is crucial in explaining how investors should choose funds and develop their investment strategies.

The performance persistence of mutual funds is appreciated as a means of selecting funds that can beat the market. Many previous studies established that the persistent performance of investment funds managers is the result of two skills: market and timing selection ability. Some authors argued that the persistent or superior performance can be achieved if managers develop both selectivity and market anticipation strategies (Oliveira et al. 2019)

The ability of fund managers to outperform the benchmark has been well documented in the literature. However, the empirical findings are mixed. Academics and researchers present two opposite points of view. The first group have argued that the conventional funds outperform stock market (Bialkowski and Otten 2011; Kiymaz 2015). The second group have showed that the mutual funds cannot outperform their benchmark over long periods without this being attributed to the luck (Ramesh and Dhume 2014). The conclusions differ depending the data selected or country selected and the time

horizon used, long or short term horizon. Additionally, many researchers found a significant difference between the performance persistence of conventional funds and religious mutual funds (Islamic, Jewish or Christian), ethical funds whether based on social responsibility principles; environmental considerations, good corporate governance. That is what caught our attention for investigating about non-conventional funds.

Over the last couple of years, investing in religious funds as well as the Islamic mutual funds have grown considerably around the world. The Islamic mutual funds can be seen as an alternative for investors used to invest in conventional funds. Islamic funds differ from traditional funds because they do not invest in companies whose operations, products or capital structure are contrary to Islamic law, as well as banking activity associated with conventional banking and all services for gambling, alcoholic beverages. Any investor, both Muslim and non-Muslim, are able to invest in Islamic mutual funds. Some investors have religious reasons and some others may regard these funds as socially responsible products. Other investors may invest in Islamic funds for diversification goals.

The increasingly growth of Islamic mutual funds attracted attention from academicians. Girard and Hassan (2008), Ashraf (2013), Mohamad and Ashraf (2015) and Omri et al. (2018) has focused on the performance of Islamic funds, and has compared these funds with their conventional counterparts across different countries. Mansor and Bhatti (2011) and Omri et al. (2018) showed that the Islamic funds are able to perform better than the conventional funds in the long term due to lower market risk globally, inferior cash outflows, lower volatility and investors' commitment to the funds.

Hayat and Kraeussl (2011) found that Islamic funds underperformed compared to the conventional funds in either bullish or bearish periods. These authors suggested that the worst performance of Islamic funds is due to poor management capacities of Islamic funds managers (as well as the market timing and stock selection abilities). However, many empirical studies (Hussein 2005; Girard and Hassan 2008) showed no difference in performance between the Islamic funds and their conventional counterparts.

The most empirical research that investigated about Islamic mutual funds still do not provide a definite answer to this most critical question: are the management ability of manager in Islamic mutual funds different from the conventional fund?

Globally, this insufficiency motivated us to compare the performance persistence and management capacities of Islamic mutual fund and conventional funds in Saudi Arabia. We are motivated to study Islamic funds in Saudi Arabia for several reasons. First, Saudi Arabia remains the key home country contributing to the biggest market share of the largest Islamic funds industry. Moreover, Saudi Islamic and other non-Islamic mutual funds have grown considerably. We are going to make a comparative study between this two funds category. Given the lack of information on the period studied for most data from Saudi funds, we had to choose a fund and how to present complete information spanning from 2011 to 2018. For this reason, in the present study, we selected the funds managed by HSBC Saudi Arabia. HSBC is a very experienced financial institution and has a good reputation managing funds in the world. Moreover, HSBC funds are well balanced and they can be divided into many sub-groups, namely, Local, International, Islamic and Conventional, consisting of the specific aim of the our study. The funds managed by HSBC are diversified in terms of investment goal classification, geographical focus and security type, thus, these different category funds give us a general view of some Saudi funds operating during 2008–2018.

This studies add to the existing literature by discussing the interaction between selectivity and market timing abilities of HSBC Saudi Islamic and conventional mutual funds. To contribute in the empirical work, we used the measure detailed by Ferson and Schadt (1996) and applied different information variables. To our knowledge, no studies have thus far used this method to assess Saudi mutual funds. An additional contribution to the existing literature is the analysis of the persistence of fund returns enables to help investor to draw conclusions about performance stability, the active managers' skill and help to choose the best funds that will outperform the market index.

The rest of the paper is organized as follows. Section 2 provides a review of the previous study focused on conventional and Islamic mutual funds. Section 3 explains empirical methodology and data

choice. Section 4 provides empirical findings on market timing and selection ability of conventional and Islamic fund managers and discusses the results. Finally, Section 5 concludes.

## 2. Literature Revue

### 2.1. Active Management Strategies

Active management seeks to take advantage of potential market inefficiencies, with the aim of maximizing returns and minimizing risk, through different strategies such as Stock Picking (Selectivity of Securities) and Market Timing (Market Anticipation). This active management strategies aims to outperform the market, by taking active positions against a benchmark.

Several studies as well as Sharpe (1966) and Wermers (1999) have showed that the performance of the mutual funds is not greater than the normal average performance, and they have suggested that the mutual funds cannot beat the market, and they did not select the winning stocks and anticipated the tendency of the market.

In the same way, Treynor and Mazuy (1966) proposed a quadratic regression analysis method to measure timing ability of the 57 fund managers over the period 1983 to 1995. They showed only one fund, which had statistically significant market timing ability. The 56 other funds have no significant evidence of timing ability. These authors did not find evidence that fund managers can outperform the market. The work of Treynor and Mazuy (1966) made significant contributions to the literature on performance-based portfolio risk-return evaluation, which distinguishes a manager's abilities to act to information and the abilities to predict systematic risk premiums and adjust the portfolio. Their study has generated much more other literature.

Using the model proposed by Treynor and Mazuy (1966), Bollen and Busse (2001) measured the performance of 230 mutual funds. In this work, the authors used both the monthly and daily data. Bollen and Busse (2001) found that in shorter periods of analysis mutual funds present a higher market timing abilities. The conclusions of this research consisted of drawing attention to an important point of view. The high funds returns generate a cash inflow that could pose a challenge in terms of investment to achieve similar rates of return. This indicates that investor cash flow dynamics can negatively impact mutual fund performance. Thereafter, Bollen and Busse (2001) explained that data frequency can play an important role in detecting managers' abilities. The authors revealed that when they use the daily data, the results showed more funds exhibit timing ability than monthly data results. This finding may be due to market exposure decisions are probably taken shorter term for many funds. Apart from pioneering studies on the active management strategies, more recent literature Ramesh and Dhume (2014) confirmed the conclusion of Sharpe (1966), which suggests that the mutual funds cannot beat the market index. Conversely, Kiymaz (2015) and Rao et al. (2017) evaluated the performance of fund managers and found a significant ability to anticipate the market. Kiymaz (2015) showed that a manager who correctly forecasts the difference between the expected return of a diversified portfolio and the risk-free rate each year tends to outperform a naïve holder. Kiymaz (2015) showed that funds that invest in risky assets, with profitability above the risk-free rate, record a positive annual gap in favor of a perfect Market Timing. Rao et al. (2017) reported that Chinese funds can successfully outperform the market index and that fund managers have a positive market timing ability. They also found that unlike developed markets, Chinese funds do not have a persistence performance.

Apart from timing skills, the performance of fund managers also depends on the stock picking skills. The approach by Ferson and Schadt (1996) has incorporated publicly available information, and has attracted the attention of academics and researchers. This approach modifies traditional performance measures (timing and stock selection capacity) using the information variables such as interest rate and market dividend yield. Ferson and Schadt (1996) incorporated conditional models into the investigation of mutual fund performance selection and timing ability for 67 mutual funds. They found that the public information contained in this model control the biases that existed in traditional models, regarding market timing and stock picking ability.

Silva et al. (2003) used the conditional measure of Ferson and Schadt (1996) on European mutual funds. They provided that using conditional measures gives clearer results than unconditional measures. According to these researchers, the European funds are generally unable to beat their benchmark. Elton et al. (2012), using the same model, found a slight presence of market timing. Clare et al. (2016) measured the performance and management capabilities of US, UK and Canadian fund managers using the same measure as Ferson and Schadt (1996). Overall, whether they applied a yield-based method or a farm-based test method, their results showed a lower capacity of funds to outperform the benchmark and to anticipate market change.

Christopherson et al. (1999) chose two information variables, the dividend yield (dividend/price), and the lagged level of short-term treasury bill (TB) yields. These authors suggest that each variable is evaluated by its expected (average) value. These measurements can be applied to single-factor models, such as the Capital Asset Pricing Model (CAPMF), or multi-factor models or to Market Timing (Treynor-Mazuy) measures. The authors suggested that investors should evaluate the manager's return series using public information.

Coggins et al. (2009) measured the performance of mutual funds using a daily data with a bivariate GARCH framework, which in particular makes it possible to condition the beta and the specific risk of mutual funds. They found similar results to those of Ferson and Schadt (1996). The conditional alpha and global performances with GARCH are considerably better than those estimated with other parameters and they persist over time. The results will be justified by the existence of a negative relation between the conditional beta of mutual funds and the market premium. Ferson and Warther (1996) provided two explanations for this negative relationship (Marhfor 2016). Firstly, this correlation can be explained by the emergence of new players, and therefore, new cash flows that are likely to change the composition of mutual funds and consequently to modify their exposure to risk. Secondly, the negative relationship may be related to the impact of transformation in the risk measures of the securities that make up the funds.

Other researchers used different measures to evaluate performance and detected the management abilities. To estimate the performance of South African equity funds and to determine the relationship between the independent and dependent variables, Tan (2015) used a multiple regression analysis (Sharpe ratio (Sharpe 1966), Treynor ratio (Treynor 1965), Jensen's alpha (Jensen 1967) methods, Treynor and Mazuy (1966) and Henriksson and Merton (1981)). The author suggested that South African fund managers could not display a good performance both in market timing abilities and selection skills.

For examining the risk-adjusted performance and selectivity skills of Indian mutual funds, Sharma and Verma (2018) used divers models as well as Sharpe Ratio, Treynor Ratio, Jensen Measure and Fama Measures. They found that only the Fama measures indicated the positive selectivity performance of Indian mutual funds over a period of 2008 to 2018.

For investigating the effects of information asymmetry on market timing in the mutual fund industry, Tchamyou et al. (Tchamyou et al. 2018) used a panel of 1488 active mutual funds for during the period of 2004–2013, and based on endogeneity-robust Difference and System Generalized Method of Moments. The dataset is decomposed into five market fundamentals (i) equity, (ii) fixed income, (iii) allocation, (iv) alternative and (v) tax preferred mutual funds. They made several conclusions. The authors documented that funds with low-grade risk exposure can outperform their benchmarks with superior exposure to fluctuation in market conditions. Agarwal and Pradhan (2018) examined the existence of superior performance of Indian mutual funds in India with several models, using unconditional models of Treynor and Mazuy (1966) and Henriksson and Merton (1981). They extended these models by incorporating Fama–French–Carhart-factors-based models. They showed evidence of selectivity and timing abilities in Indian open-ended fund managers. According to these authors the explanation of their results is that the selection of titles takes place much more frequently than at monthly intervals. The absolute superior returns are lower for multifactor models. Pilbeam and Preston (2019) studied the performance of 355 actively managed Japanese Mutual Funds between 2011 and 2016. Using Jensen's alpha measures, the authors found that the Japanese Mutual Funds

underperformed the benchmark. When the Treynor and Mazuy measure were used, the results showed that only 33 funds have significant positive market timing ability which was largely offset by 31 funds with significant negative timing ability.

*2.2. Persistent Performance*

The evaluation of active management takes into account of the persistence of funds' abnormal returns relative to their benchmarks; if there is low persistence, the abnormal performance are probably motivated by luck rather than skill.

Before the work of Carhart (1997) the majority of studies had reported that past performance can weakly predict future performance. However, Carhart (1997) found evidence of persistence performance in American mutual funds. Carhart (1997) included the 'Momentum' variable, which refers to the risk premium on the past performance. This factor represents the difference between the returns of the portfolios with a high historical return and the returns of the portfolios with a low historical return. Indeed, Carhart (1997) indicated that there managers are not able to maintain a good performance without taking more risk. This study further suggested that underperforming stocks and those that perform very well over the last six months retain the same trends in the future. Carhart (1997) showed that funds with the worst recent performance continued to offer terrible returns, which momentum and expenses could not fully explain. Differences in style, luck, skill and cost orientation determine the relative performance of the funds. Therefore, the performance gives an idea about managerial skills. However, there is no good skill in creating long-term performance persistence, as differences in style and expense may also create this effect. Funds with long-term positive past experience tend to have lower fees than losers, and these cost differences are likely to persist.

Working on 18 stock markets in Asia, Latin America and Eastern Europe, Cakici et al. (2013) used Carhart (1997) four-factor model to evaluate persistence performance of mutual funds. They concluded that persistence is more significant in smaller companies.

Since the work by Carhart (1997), several studies has been done on persistence in the performance of mutual funds, but the finding is mixed. Bollen and Busse (2001) documented that the Mutual funds with a past higher (lower) performance tend to offer higher (lower) performance in the future. However, Bialkowski and Otten (2011), Brown and Goetzmann (1995) found evidence of predictability in the performance of mutual funds over short time horizons. Whereas, Grinblatt and Titman (1992) showed that the performance of mutual fund persist only in the long term. Brown and Goetzmann (1995) confirmed that the persistence of performance depends on the period under study.

However, Carhart (1997) provide evidence that the effect of "hot hands" is due to the persistence of expenditure ratios and the adoption of stability strategies. Papadamou and Siriopoulos (2004) showed that the performance of US funds investing in Europe is lower than the benchmark (Eurostoxx), and in the most case, the phenomenon of "hot hands" does not persist. Mutual funds that have performed well over the past five months will continue to outperform over the next four months.

Papadamou et al. (2017) investigated how mutual funds performed in Japan before and after the 2008 outburst of the global financial crisis. The authors showed that active management over the two periods does not add significant value to the mutual fund performance. Thus, no funds could outperform the market index.

More recently, Mateus et al. (2019) performed research using the 817 active UK long-only equity mutual funds during the period 1992–2016. They documented that the funds is with the most significant positive alpha that was already recorded a good performance in the previous year. In terms of parametric and non-parametric measurements, these funds showed a persistence of performance.

By contrast, several authors did not find persistence in the performance of mutual funds over time (see Christensen 2005; Huang and Mahieu 2012; Rao et al. 2016, 2017). Using parametric and non-parametric methodologies, Christensen (2005), evaluated performance persistence of 47 Danish mutual funds. They concluded that in general Danish mutual funds perform neutrally, returns are non-persistent and funds don't have the timing and selectivity skills.

Rao et al. (2017) evaluated the equity funds in China. Appling the capital asset pricing model (CAPM) and the Carhart four-factor model, the authors showed that the equity funds outperform their benchmark but they do not find any evidence of persistence in fund performance.

*2.3. Previous Literature on Islamic Mutual Funds*

Islamic funds are in full compliance with Sharia, which formally prohibits investing in companies engaged in noncompliant operations. In order to fulfill with the needs of investors who wish to comply with the Sharia rules, the specialists have relaxed the ideal form of Sharia investments and formulated a relatively more balanced investment criterion, known as Sharia-compliant equity investment guidelines.

The diversity of Sharia selection criteria extends the selection choices of Sharia candidates. However, there are several problems with the investor who prefers Islamic investment. First, the Sharia investor always has specific criteria with the constraint of the opportunity cost. Secondly, there is currently no mutual agreement on the use of financial screens.

Mansor and Bhatti (2016) argued that the goal of the Islamic mutual funds is to meet ethical and religious objectives without harming the conventional needs in terms of liquidity, diversification and return. Conversely, according to other authors (Boo et al. 2017), Islamic mutual funds are a means of hedging against financial crises, because they invest in higher liquid and low-leverage stocks, likely to outperform the conventional benchmarks.

The most research in Islamic finance have always been interested in investigating the question whether the strategies pursued by Islamic funds help them to beat the market index and to perform better their conventional counterparts in terms of performance and active management strategies.

Recently, research on Islamic Mutual Funds has grown considerably and gained momentum in the global market. Several studies concentrated on risk and return behavior comparisons between conventional and Islamic mutual funds and found quiet mixed results. Some of them did not find any significant difference between performance Islamic fund and the performance of their counterparts. Others studies have shown that these funds record performances and totally different management styles. As a whole, we can say that the empirical literature on Islamic mutual funds performance analysis is still in its infancy. Girard and Hassan (2008) used different performance measures for compare five FTSE (Financial Times Stock Exchange) Islamic indices and five conventional benchmarks (MSCI) (Morgan Stanley Capital International). They did not find a significant performance differential or managerial capabilities between Islamic and conventional funds (Dow Jones Islamic indices and Morgan and Stanley conventional indices). Consistent with other studies, the study of Hassan and Girard (2010) produced similar evidence to that of Girard and Hassan (2008). They concluded that there was no difference in performance between Islamic and traditional funds in Malaysia. Al Rahahleh and Bhatti (2017) compared Islamic and conventional funds in Saudi Arabia and they found that Sharia compliant funds did not outperform or underperform their benchmark (the S&P Saudi Arabia Domestic Sharia Index).

On the other hand, other studies such as Hussein (2005) have shown that compliance with Sharia law has a negative impact on fund performance during downturns. Other researchers explain this underperformance of Islamic funds compared to traditional funds. They suggested that this underperformance is due to the many restrictions imposed by Sharia. Merdad et al. (2010) studied the performance of Saudi funds managed by HSBC. They found that Islamic funds underperformed conventional funds during overall and bullish periods, but outperformed during periods of economic downturn and financial crisis. In addition, they found that Islamic HSBC funds managers have a good ability of market timing and selectivity skills only during the downturn and the conventional funds have a significant ability of market timing and selectivity skills only during the bullish period. Hayat and Kraeussl (2011) found that Islamic funds have performed worst in either economic scenario, bullish or bearish period. They further suggested that managers of Islamic funds exhibit poor stock selection and market timing abilities. Rana and Akther (2015), by using different performance measures

(Sharp ratio, Jensen alpha, Treynor ratio), investigated the Pakistani Islamic and conventional funds. They employ The GARCH-M framework to investigate about risk-return trade-off in the context of both Islamic and conventional stock indices. They found that the Islamic stock index underperform their conventional counterparts due to availability of a smaller investment universe, increased monitoring costs and limited diversification. On the other hand, other researchers believe that Islamic investment offers a good economic and social sense to evaluate potential investments. Islamic ethical investors can side with their potential investments and their religious and ethical beliefs, which will not only give them peace of mind, but also a legal monetary reward (Rana and Akther 2015). Mansor and Bhatti (2016) found that IMF and CMF funds are able to outperform market performance throughout the overall study period. However, they showed that during financial crises and pre-crisis periods; Islamic funds did not outperform or underperform conventional funds. This study concluded that the performance of ethical funds is more persistent, especially during and after the crisis period. Elfakhani et al. (2007) argued that the ability to outperformance or underperformance the market depends on the measure and benchmark used for the performance evaluation.

According to the empirical literature the excess performance of investment funds had been attributed to management skills and to selection criteria if an ability inherent in selecting the right securities (selection) and choosing the best timing. The existing literature focuses primarily on determining abnormal performance (stock selection and market anticipation) using models such as Treynor and Mazuy (1966) and Jensen (1967). Mohamad and Ashraf (2015) suggested that the stock selection and rebalancing of funds to comply with Islamic laws may result in superior returns for the investing public. The results between the Islamic funds from developed markets (North America and Europe) and those from emerging markets are totally different. Mohamad and Ashraf (2015) suggested that Sharia law helps the Islamic funds to select the ideal stocks for example, the stocks of firm who are not financially distressed, are growth oriented and are exhibiting a positive momentum. However, with using both parametric and non-parametric based tests, their results indicated that the Islamic funds showed a negative significant market timing ability. The authors argued that Islamic indices do not necessarily have a higher risk or a lower diversification than conventional indices. The difference in performance is most pronounced in developed regions where leverage plays a key role in companies' capital structure. In addition, the importance of growth and momentum factors is more evident among the S&P Dow Jones Islamic index that use the market value of stocks as a denominator in the calculation of financial filter ratios. There is not a big difference between emerging market Islamic index returns and conventional indices.

Abdullah et al. (2007) studied the performance of Islamic and conventional unit trust funds in Malaysia. They used the Treynor and Mazuy (1966) model and found that Islamic funds performed better than conventional funds only during bearish period. However, during the bullish period, conventional funds outperformed the Islamic funds. They concluded that there was little evidence of market timing and stock picking ability in Malaysian mutual funds.

Ashraf (2013) found that Saudi IMFs perform better than conventional funds from 2007 to 2011. Managers of Islamic mutual funds have a better selection ability than managers of conventional funds. However, in Islamic mutual funds, the managers do not have superior market timing capabilities. In contrast, managers of conventional funds not only exhibit a poor stock selection ability but also exhibit significant negative market timing ability. The combination of superior stock selection ability of Islamic mutual funds and the negative market timing ability of conventional funds suggests that Islamic mutual funds offer better hedging opportunity to investors during periods of economic downturn.

Concentrated on only after financial crisis, Omri et al. (2018) compared the risk-adjusted performance and investment style of Islamic mutual funds and conventional funds. They applied models of absolute and relative risk-adjusted measures with single factor (Jensen) and multifactor (Carhart). To obtain diverse portfolios, they used geographical and religious diversification of the sample. Their results showed that the Islamic portfolios were able to outperform the conventional portfolios given similar risk exposure, and produced comparable results under lower market risk

globally. Omri et al. (2018) concluded that respect for Sharia did not have a negative impact on the performance of Islamic funds and did not entail any additional risk for holding such investments. This suggests that Islamic funds in Saudi Arabia are very competitive, they take reasonable risks and generate respectable returns.

Using the performance measurement approaches of Carhart (1997) and Ferson and Schadt (1996), Bauer et al. (2007) provide evidence that risk-adjusted returns between conventional and ethical Canadian mutual funds do not differ. Moreover, Chau et al. (2003) found that management skills do not extremely matter, and they showed no significant differences between styles.

### 2.4. A Contextual Overview of the Saudian Mutual Fund Industry

This paper focuses the critical investigation of Islamic Saudi mutual funds. Saudi Arabia is an interesting context for further investigating the performance and active management for several reasons. Firstly, Saudi Arabia is still the biggest market for mutual funds in GCC countries and accounts for more than 80% of the total funds market in GCC region (Ernst and Young 2015). In 2017 in Saudi Arabia, there exist more than 275 mutual funds managed by more than 40 fund managers. Secondly, it is considered one of the few countries that strictly adhere to Sharia rules and follow regulations derived from Islamic law. Moreover, more than 70% of the mutual funds operating in Saudi Arabia are Sharia compliant, compared with just 21% in the rest of the GCC region (Ernst and Young 2015). This means that the performance and management of these funds can be different from their conventional counterparts and depends essentially on fund geographical focus. Thirdly, the Saudi mutual fund industry has been growing considerably in the last couple of years. According to the official web of Saudi stock market TADAWUL, in 2015 Saudi equity mutual funds based on geographical location present 46% of locally mutual funds, 34% internationally focused and 20% are invested in the Arab or GCC countries. According to Saudi Arabian Monetary Authority (SAMA), the total value of assets under management (AUM) of Saudi funds augmented by 25.5 percent year-on-year (YoY) by the end of 2017. The Saudi market is intensively participates in activities aimed at strengthening financial liberalization. This market continually attempts to improve the efficiency of the market and the corporate governance of companies; however, it suffers from some limitations existing in emerging markets, such as lack of complete and free information, transaction costs and illiquidity of stocks (El-Masry and El-Mosallamy 2016). To the best of our knowledge, some research on Saudi Arabia based IMFs performances is available, but little work has been done to provide a comprehensive managers ability to beat a market and to have a good and persistent performance and analysis of both IMFs and conventional funds offered in Saudi Arabia.

## 3. Data and Estimation Methodology

### 3.1. Sample and Data

The objective of the present study is to following and to expanding the work of Merdad et al. (2010). The previous study of Merdad et al. (2010) investigated the impact of Sharia screening on the performance of HSBC mutual funds. However, this paper expanded their results by examining the selectivity and market timing abilities among Islamic and conventional HSBC funds.

For this study, our data consist of 15 mutual funds managed by HSBC spanning from 2011 to 2018. Taking into account the survivor bias, we excluded those portfolios that did not have a complete historical data from April 2011 until December 2018. We also considered in our study the population that includes only the surviving funds on our study period and excludes from our sample those who were present at the beginning of sample period, but who did not remain until the end. We then eliminated from the total HSBC-managed funds those that were closed before 2018 and those created after 2011, hence, obtaining the 15 funds.

We calculated the monthly returns of Saudi mutual funds from their net asset value that have been provided by the official site of the Saudi Stock Exchange (Tadawul) (www.tadawul.com.sa), the official site of HSBC Saudi Arabia Limited (www.hsbcsaudi.com) and the Zawya database. Moreover, the funds

management by HSBC can be divided into five sub-groups, Local, Arab, Global, Islamic and Conventional, consisting of the particular aims of this study. Based on their type and geographical focus, these 15 Saudi mutual funds are grouped into equally balanced portfolios and divided into five subgroups based on geographical focused and Sharia compliance: Local/Islamic, local/conventional, global/Islamic and global/conventional, and Arabic Islamic. We did not find any Arab conventional fund managed by HSBC between 2011 and 2018. All these portfolios are denominated in Saudi currency (RAS) and USD converted into a single currency, USD.

For more correct and unbiased results, each portfolio was compared to the conventional and Islamic indices that have the same geographical focus as the examined portfolio. The selected funds and the used benchmarks are given in Table 1. We employed six different market indices. These indices diversified under two main classifications. The first classification is the Islamic indices: (1) S&P Quality Saudi Arabia Sharia Index, (2) S&P Pan Arab Composite Sharia, (3) S&P Global BMI Sharia. The second classification is the conventional indices: (1) S&P Pan Arab Composite S&P Saudi, (2) Arabia total return index and (3) S&P GLOBAL 1200.

**Table 1.** Descriptive portfolios and benchmarks.

| Portfolios | Funds Benchmarks |
| --- | --- |
| LS: Local Sharia | LSBM: S&P Quality Saudi Arabia Sharia Index |
| LC: Local Conventional | LCBM: S&P Saudi Arabia Total Return Index: |
| AS: Arab Sharia | ASBM: S&P Pan Arab Composite Sharia: |
| IC: International Conventional | ACBM: S &P Pan Arab Composite: |
| IS: International Sharia | INSBM: S&P GLOBAL 1200: |
|  | INCBM: S&P Global BMI Sharia: |

Note: The table presents different five open-ended mutual fund portfolios managed by HSBC over the period April 2011 to December 2018. We are divide the funds managed by HSBC into three geographical focus (Arab-local-international) and Sharia compliance focus (Islamic–conventional).

## 3.2. Methodology

In order to evaluate the performance of the 15 HSBC funds and to assess the ability of their managers to outperform the market through selectivity and market anticipation, we used the model of Ferson and Schadt (1996). Each portfolio was initially examined against two market indices, the Islamic and the conventional indices.

We chose the model proposed by Ferson and Schadt (1996) for several reasons. Firstly, this model takes into account the manager's ability to select and anticipate both the best stocks. Secondly, the method is able to estimate the persistence of the performance of these funds over time.

The finding of Ferson and Schadt (1996) are very relevant to our research. First, they found that the level of risk to which funds are exposed varies in response to public information. Second, they observed a significant difference between conditionally measured and traditionally measured performance.

Based on a daily publication of the Saudi stock exchange market, we have monthly adjusted the net asset value (NAV) that will help us to calculate the $R_{p,t}$:

$$R_{p,t} = \frac{(NAV_{it} - NAV_{it-1}) + D_t}{NAV_{it-1}} R_{it} \tag{1}$$

where:

$R_{i,t}$: The return of the funds i in the month $t$.
$NAV_{it}$ and $NAV_{it-1}$: Net asset values of the fund $i$, respectively, at the end of the month $t$ and $t-1$.
$D_t$: Dividend paid out for Fund $i$ at month $t$.

$$R_{it} = 1/n \sum_{i-1}^{n} R_{i,t} \tag{2}$$

$R_{it}$ : averege return of the portfolio p during the month $t$.

Therefore, the model by Ferson and Schadt (1996) was constructed as follows:

$$R_{p,t} = \alpha_p + \beta_1 R_{m,t} + \beta_2 R_{m,t}^2 + \beta_3 (Zt_{-1}R_{m,t}) + \varepsilon_{p,\,t} \tag{3}$$

where:

$R_{P,t}$ represents the average rate of return on portfolio "p" in month $t$. $R_{m,t}$ is the average market Benchmark.

$\alpha_p$ and $\beta_1$ are the coefficient indicate the stock selection, and the systematic risk. If the $\alpha$ is positive, the fund manager has an ability to forecast security prices. On the other hand, if the $\alpha$ is negative, the manager of the mutual funds is not doing well in funds selection.

$\beta_2$ is the market timing ability. If $\beta_2 > 0$ and is significant, the fund manager has an ability to beat the market. $\beta_3$ captures the response of the fund's managers to public information ($Z_{t-1}$).

$\beta_3(Z_{t-1}R_{m,t})$ controls the public information effects.

$\varepsilon_{p,\,t}$ The residual excess return of the portfolio p.

Our model is different from that of Ferson and Schadt (1996) because validation will focus only on two information variables, while Ferson and Schadt (1996) introduced four information variables.

Indeed, the main difficulty when using this model will be to determine the information vector that will be used. We chose the variable Ramadan effect and the variable treasury bills at three months, as an information variable. Regarding our choice the information variable, first, the value of the lagged level of short-term treasury bill (TB) yields is similar to the study for Christopherson et al. (1999) and Ferson and Schadt (1996). Second, the choice of Ramadan effect as well as a dummy variable is motivated by the fact that this event could have an influence on the performance, behavior and abilities of the agent who manage the funds in a Muslim country.

Ferson and Schadt (1996) have shown that the model estimated by OLS is equivalent to an estimate by the method of Hansen (1982). For our case we will estimate the model of Ferson and Schadt (1996) by following the GARCH (1.1) process for each portfolio. In order to assess the Saudi manager market capacity we use:

$$R_{p,t} = \alpha_p + \beta_1 R_{m,t} + \beta_2 R_{m,t}^2 + \beta_3 (T_{Bt-52}R_{m,t}) + \beta_4 (D_{ramadan} R_{m,t}) + \varepsilon_{p,\,t} \tag{4}$$

If $\beta_2 > 0$ and significant then the fund manager has a capacity to beat the market.

$\beta_3$ and $\beta_4$ indicating the response of the fund's managers to public information.

$T_{Bt-52}$ is the return of three month treasury bills for the previous years.

$D_{ramadan}$ is a dummy variable captures the Ramadan effect on return. where $D_{ramadan} = 1$ if it is Ramadan month or zero otherwise.

*3.3. Hypothesis*

The aims of this study were twofold. Firstly, identify the investment strategy of Islamic and conventional mutual funds managed by HSBC and the ability of Islamic portfolio and conventional portfolios managers to anticipate market movements and select the best stocks at the right time. Secondly, this research addresses the persistence of the different portfolios. Based on the reviewed theoretical and empirical literature on Saudi Arabia and Mutual funds, we investigate the following hypothesis.

**Hypothesis 1.** *Both Islamic and conventional portfolios reveal similar market timing and selectivity skills on the local and global and Arabic basis.*

**Hypothesis 2.** *The market ability (selectivity and timing) of Islamic portfolios manager are more sensitive to Ramadan effect than conventional portfolios, on locally, globally and Arabic bases.*

**Hypothesis 3.** *Both Islamic and convention portfolios perform persistently.*

*3.4. Descriptive Statistics*

Table 2 presents the summary statistics on HSBC Islamic and Conventional mutual funds and the associated benchmarks based on the monthly aggregated return funds from April 2011 to December 2018. The Islamic and Conventional fund returns are calculated based on an equally weighted portfolio of all funds belong to each of the portfolios. Panel A reports the descriptive statistic for five portfolio divided into sub-groups, Local, Arab and Global; each group was also divided into both Islamic and non-Islamic portfolio. Panel B reports the same descriptive statistic for six market indices: S&P Quality Saudi Arabia Sharia Index (locally focused on Islamic index), S&P Saudi Arabia total return index (locally focused on conventional index), S&P Pan Arab Composite Sharia (Arab focused on Islamic index), S&P Pan Arab Composite (Arab focused on conventional index), S&P GLOBAL 1200 (international focused on conventional index) and S&P Global BMI Sharia 1200 (international focused on Sharia index).

**Table 2.** Summary statistics on return performance on Islamic and conventional HSBC funds.

| Portfolio | MEAN | STDV | MIN | MAX | MEDIAN |
|---|---|---|---|---|---|
| Local Sharia (LS) | 0.0333 | 5.52 | −34.9850 | 22.107 | 0.1063 |
| Local Conventional(LC) | 0.3143 | 6.1713 | −20.7901 | 13.314 | 0.4755 |
| Arab Sharia (AS) | −0.0361 | 1.4028 | −8.8396 | 8.8043 | −0.0114 |
| Inter Sharia (IS) | 0.2084 | 2.9184 | −9.2039 | 5.5153 | 0.6625 |
| Inter Conventional (IC) | 0.05897 | 0.16482 | −0.4185 | 0.6269 | 0.0054 |
| **BENCHMARK** | **MEAN** | **STDV** | **MIN** | **MAX** | **MEDIAN** |
| S&P Quality Saudi Arabia Sharia Index (ISBM) | 0.6847 | 5.1955 | −18.6807 | 16.4436 | 0.6525 |
| S&P Saudi Arabia Total Return Index (LCBM) | 0.4968 | 5.9591 | −18.7462 | 18.6326 | 1.1004 |
| S&P Pan Arab Composite Sharia (ASBM) | 0.2582 | 4.6255 | −18.6213 | 18.9324 | 0.2855 |
| S &P Pan Arab Composite (ACBM) | 0.1511 | 3.4526 | −11.1467 | 8.2988 | 0.4434 |
| S&P GLOBAL 1200 (ISBM) | 0.4958 | 3.6368 | −10.8828 | 10.3697 | 1.1637 |
| S&P Global BMI Sharia (ICBM) | 0.5672 | 3.4808 | −9.0738 | 10.0082 | 1.2564 |

Note: This table provides the descriptive statistics on the sample of the five equally weighted portfolios managed by HSBC funds. These portfolios are formed based on the following characteristics of the funds geographical focus (local, Arab and conventional) funds and Sharia compliancy (Islamic and conventional). The model is significant at 10%, 5% and 1%, respectively.

According to this table, we can show some preliminary findings. First, the monthly returns of the portfolios managed by HSBC over a period of eight years are significant. We note that all portfolios have a positive average return with the exception of the Arab Sharia portfolio. It is clear that the portfolios studied achieve significantly lower averages than corresponding benchmarks. However, those who achieved the highest average returns over the period of analysis are local portfolios, either Islamic or conventional portfolios. Indeed, these two categories base the exception and record a higher standard deviation than their benchmarks. Table 2 also documents the local conventional funds have higher performance with a higher risk that local Sharia portfolio. However, the international Sharia portfolio has higher performance with a higher risk that international conventional portfolio. The benchmarks showed a similar pattern. All international, local and Arab portfolios underperformed their corresponding benchmarks with a smaller mean.

Our results seem to contradict those of Omri et al. (2018), according to whom the Islamic funds outperformed their conventional counterparts. However, our results agree with Merdad et al. (2010), who found that during the full analysis period the Saudi Islamic mutual funds managed by HSBC unperformed than their conventional counterparts.

Following Merdad et al. (2010) and Omri et al. (2018), as a second step, we calculated the absolute performance difference between Islamic and conventional portfolios and their index. We found that the differences mean between portfolios and different benchmarks are not statistically significant (Table 3).

**Table 3.** Mean difference.

|      |      | MEAN DIFF | T-STAT | F-TRST | P-VALUE |
|------|------|-----------|--------|--------|---------|
| LS   | LC   | −0.2843   | 0.6330 | 0.0350 | 0.5509  |
| LS   | ISBM | −0.6515   | 0.4602 | 0.0983 | 0.4597  |
| LS   | LCBM | −0.4660   | 0.5968 | 0.0000 | 0.6198  |
| LC   | ISBM | −0.3657   | 0.6689 | 0.6461 | 0.9096  |
| LC   | LCBM | −0.1820   | 0.8441 | 0.0000 | 0.2470  |
| ISBM | LCBM | 0.1840    | 0.8110 | 0.0000 | 0.1689  |
| AS   | ASBM | −0.2480   | 0.5875 | 0.0000 | 0.5583  |
| AS   | ACBM | −0.0990   | 0.4855 | 0.0000 | 0.6285  |
| ASBM | ACBM | 0.2660    | 0.1791 | 0.0054 | 0.8583  |
| IS   | IC   | 0.2211    | 0.4979 | 0.0000 | 0.6198  |
| IC   | ISBM | −0.4361   | 1.1688 | 0.0000 | 0.2455  |
| IC   | ICBM | −0.5081   | 1.3867 | 0.0000 | 0.1689  |
| IS   | ISBM | −0.2870   | 0.5986 | 0.0350 | 0.5509  |
| IS   | ICBM | −0.3590   | 0.7424 | 0.0983 | 0.4597  |
| ISBM | ICBM | −0.0720   | 0.1139 | 0.6461 | 0.9096  |

In order to check the dependence between the different portfolios as well as the portfolio and their benchmarks, at the same time, we did a correlation analysis. According to Table 4, we found that most Islamic and conventional and their benchmarks have statistically significant positive relationships overall. These results confirmed the findings of Mansor and Bhatti (2011) and Omri et al. (2018). However, the International Islamic benchmarks (ISBM) and the International conventional benchmarks (ICBM) were strongly positively correlated with each other. We can conclude that this combination is almost a perfect correlation.

**Table 4.** Person correlation.

| | Panel 1 | | | |
|---|---|---|---|---|
| | Local | | | |
| Portfolio | LS | LC | LSBM | LCBM |
| LS   | 1.00   |       |      |      |
| LC   | 0.54   | 1.00  |      |      |
| LSBM | −0.092 | −0.26 | 1.00 |      |
| LCBM | −0.027 | −0.09 | 0.08 | 1.00 |
| | Panel 2 | | | |
| | International | | | |
| Portfolio | IS | IC | ISBM | ICBM |
| IS   | 1      |        |       |   |
| IC   | −0.043 | 1      |       |   |
| ISBM | 0.555  | −0.026 | 1     |   |
| ICBM | 0.503  | 0.0438 | 0.927 | 1 |

Note: The model is significant at 10%, 5% and 1%, respectively.

## 4. Results and Discussion

### 4.1. The Market Timing and the Selection Skills of Islamic and Conventional Funds

Table 5 reports the result for the five portfolios benchmarked each time against two different market indices. Following Omri et al. (2018) the performance of each funds is regressed against the relevant benchmark. Focusing on Table 5, results indicate that 70% of funds operated by HSBC have significant statistics market selection.

**Table 5.** Regression analysis on market timing and selectivity.

| Portfolio | LS | | LC | | AS | | IC | | IS | |
|---|---|---|---|---|---|---|---|---|---|---|
| Benchmark | LIBM | LCBM | LCBM | LSBM | ASBM | ACBM | INCBM | INSBM | INSBM | INCBM |
| $\alpha$ | 1.1480 ** | 1.0053 * | 0.9791 | 0.9773 | −0.2200 *** | −0.3070 *** | 0.001 *** | 0.0070 *** | 0.2280 | 0.4180 *** |
| t($\alpha$) | 2.0733 | 1.8146 | 1.4659 | 1.5492 | −3.4122 | −4.4068 | 20.93 | 26.69 | 1.0802 | 6.635 |
| $\beta1$ | −0.0260 | 0.06140 | 0.2070 | 0.4160 | −0.0330 | 0.1310 *** | 0.000 *** | −5.780 | 1.0870 *** | 1.1410 *** |
| t ($\beta1$) | −0.090 | 0.3283 | 0.6424 | 1.1994 | −2.2475 | 3.222 | 8.879 | −0.30485 | 172.68 | 9.738 |
| $\beta2$ | −0.012 | −0.007 | −0.013 | −0.006 | 0.003 | 0.002 | 0.000 *** | 0.000 *** | −0.023 *** | −0.029 *** |
| t ($\beta2$) | −0.560 | −0.7294 | −1.1297 | −0.3194 | 0.7293 | 0.4462 | 32.767 | −6.5763 | −2.5502 | −3.7936 |
| $\beta3$ | −0.124 | −0.155 | −0.361 | −0.640 ** | −0.002 | −0.096 *** | 0.001 *** | −0.003 *** | −0.512 *** | −0.644 *** |
| t ($\beta3$) | −0.451 | −1.0545 | −1.471893 | −1.852392 | −1.0153 | −2.7967 | 8.839 | −9.2097 | −7.1719 | −8.1524 |
| $\beta4$ | −0.456 | −0.057 | 0.041 | −0.496 | 0.026 | 0.015 | 0.000 ** | | −0.108 | 0.095 |
| t ($\beta4$) | −0.513 | −0.1260 | 0.1199 | −0.7694 | 0.4929 | 0.2940 | −2.0016 | −0.8922 | −0.5304 | 0.5549 |
| $\alpha0$ | 19.083 *** | 16.809 *** | 2.804 | 1.246 | 0.192 | 0.221 *** | 0.000 | 0.000 | 0.053 | 3.376 *** |
| t ($\alpha0$) | 2.3118 | 3.2372 | 0.5877 | 0.3514 | 1.4568 | 2.773 | −0.0032 | −0.3798 | 1.0698 | 2.686 |
| $\alpha'1$ | 0.242 * | 0.329 *** | 0.054 | 0.075 | 3.601 *** | 3.459 *** | 51.024 *** | 16.575 *** | −0.086 *** | 0.650 ** |
| t ($\alpha'1$) | 1.6785 | 3.2033 | 0.5977 | 0.8979 | 3.780 | 4.084 | 9.388 | 6.3847 | −4.4875 | 2.0854 |
| $\beta'$ | −0.125 | −0.063 | 0.867 *** | 0.903 *** | 0.000 | −0.002 | 0.000 | 0.014 ** | 1.082 *** | −0.108 |
| t ($\beta'$) | −0.300 | −0.2617 | 4.1257 | 5.5738 | −0.0030 | −0.0851 | −0.1107 | 2.0492 | 29.796 | −0.9283 |
| R2 | 0.119 | 0.017 | 0.117 | 0.646 | 0.002 | 0.006 | 0.113 | 0.104 | 0.231 | 0.127 |

Note: this table presents the performance of the 15 portfolios managed by HSBC from April 2011 to December 2018. This table displays the results of the Ferson and Schadt (1996) model, where the $\alpha$ coefficient indicate the selectivity skills and the $\beta2$ coefficient indicate the market timing ability. The model is significant at * 10%, ** 5% and *** 1%.

The regression results show that the local Islamic portfolio (LS) have a statistically positive significant $\alpha$. These results report the selectivity skills for local Sharia portfolio (LS) are lower when the portfolio is benchmarked against the S&P Saudi Arabia total return index (LCBM). Therefore, when benchmarked this portfolio with the S&P Quality Saudi Arabia Sharia Index (LSBM), we showed that the local Islamic portfolio have a significant and higher selectivity skills. This conclusion holds for the two different market benchmarks. On the other hand, local conventional portfolios (LC) generally do not reflect any evidence for a managers selectivity skills even when using two different market indices.

This finding indicates that market selection skills of local Islamic portfolio (LS) are stronger than the local conventional portfolio (LC). The managers of Saudi Islamic portfolios (LS) are able to take advantage of earnings anomalies, and thus, they make the right choice among stocks and sectors of activity. Our results specify that local Islamic funds are highly competitive and generate good returns with reasonable risk. The result may indicates the presence of home bias towards Islamic assets, due to the fact that manager or investor prefer invest in Saudi Islamic companies how prohibit alcohol, pork products, Gambling and any other forms whose activities the Sharia Board feels are prejudicial to the interests of Islam or Muslims.

We observed that the HSBC managers invested locally performed better in stock selectivity skill. The Islamic funds can able to select the ideal stocks for example, the stocks of firm who are not financially distressed, are growth oriented and are exhibiting a positive momentum.

Using two different benchmarks, S&P GLOBAL 1200 (INSBM) and S&P Global BMI Sharia (INCBM), we found that the international conventional portfolio (IC) have a good capacities to forecast security prices. This same table reports that if the Arab Islamic portfolios present negative and statistically significant $\alpha$, then these managers are not doing well in funds selection.

Nevertheless, when the S&P Global benchmark Sharia or S&P GLOBAL 1200 (INSBM) are used as international market index, the results indicate that international Islamic portfolios (IS) have a higher statistically positive significant $\alpha$ than the international conventional portfolio (IC). The results also

provide evidence of better performance in stock selectivity skill of Islamic portfolios (IS) comparing to the international conventional portfolios (IC).

An overall sample reveals a significant stock selectivity of the HSBC funds' performance over the period of 2011 through 2018. Finally, 50% of the managers of these funds are able to take advantage of compensation anomalies and so they know how to make the right choice in the selection of securities and business sectors. These portfolios are able to generate higher excess returns than all the portfolios in the study.

The results of significant stock selectivity in Saudi Islamic portfolio are inconsistent with the finding of Hayat and Kraeussl (2011). The contrasting finding can be due to the shorter duration of the study period and the limited sample of Saudi mutual funds. However, the evidence of positive market selectivity of the fund managers supports the previous study by Omri et al. (2018), as they denoted that the Islamic funds are able to obtain superior return by investing in large, well-diversified corporations while applying the purification process through profits distribution.

In keeping with the approach of Ferson and Schadt (1996), positive and statistically significant β2 value indicate evidence of timing ability of the associated mutual funds. Negative significant β2 values may be interpreted as negative market timing. Table 5 reports a significant market timing ability coefficient for only 40% HSBC portfolios (as well as international funds; IC and IS).

When the international-focused conventional portfolio (IC) is benchmarked against two different Index, the results report a positive and statistically significant coefficient of timing skills at 1 percent. We conclude that on average the managers of HSBC conventional funds investing internationally have little market timing ability and selection skills. In addition, the managers of this portfolio have little capacity to beat the market index by predicting its movements and buying and selling accordingly.

However, the results show that there are differences in market timing between the Islamic and the conventional internationally-focused portfolios. We showed that Islamic internationally-focused portfolios (IS) present negative and statistically significant coefficient regardless of the choice of the benchmark portfolio. This negative value may be interpreted as the incapacity of (IS) fund manager to beat the market. Moreover, the (IS) fund managers tend to augment (reduce) their exposures to the market in question at a time when the market is declining (rising). This evidence about the poor market timing is consistent with research conducted by Fauziah et al. (2002), who failed to capture the contribution of a manager's timing activities to fund returns.

Furthermore, the results indicate evidence of differences in market timing skills between the Islamic and the conventional internationally-focused fund portfolios. The paper also provides evidence that the Conventional fund managers performing better in market timing ability (significant at 10%) in relation to the market performance. These results are consistent with the findings of research conducted by Hoepner et al. (2011) and Hayat and Kraeussl (2011). According to these authors, this insignificance of market timing can be recognized like the fact that the flow of Islamic funds is more stable than conventional funds, because on average the manager of Islamic funds select stock with long term perspectives and lower market risk.

Globally, the performance of the (IS) funds in stock selectivity skill is higher, and in market timing ability is slightly lower than its (IC) funds peers, with both statistically significant at 10%. Finally, it is evident that the investors are better off investing in international-focused conventional funds (IC) than international-focused Sharia funds (IS). This finding is due to the Sharia selection process and the way in which low-leverage firms are part of Islamic funds. This finding are inconsistent with the results of Treynor and Mazuy (1966) and Girard and Hassan (2008), who suggested that there are no differences in market timing skills between the Islamic and the conventional internationally-focused fund portfolios.

The inability to timing skills of the 60% other HSBC portfolios (LS; LC and AS) can be endorsed to the fact that the flow of local and Arab funds is more constant and persistent than HSBC international portfolios. Because investors in domestic and Arab funds are less sensitive to volatility of market returns, the manager chooses to invest in the long-term stocks.

Thus, regarding the first hypothesis, the results reveal that both Islamic and conventional portfolios have similar market timing and selectivity skills on the local and global and Arabic basis. Thus, the first hypothesis can be rejected, implying there is a statistically significant difference in terms of performance and strategy management between Islamic and conventional portfolios.

After that, we found strong evidence that the holy month of Ramadan has no affect on the market ability of HSBC portfolios, except on the international-focused conventional funds. The coefficient $\beta 4$, which reflects the response of the portfolio manager's beta to Ramadan effect is not statistically significant for 90% HSBC portfolios. According to these finding, no evidence was found to support the effect of Ramadan on the performance of the Islamic and conventional HSBC Saudi Fund.

For this reason, we can reject the second hypothesis, which was concerned with that the selectivity and market timing of HSBC Islamic portfolios are more sensitive to Ramadan effect than HSBC conventional portfolios. These results are inconsistent with the work of Mitchell et al. (2014), who suggested evidence for a significant Ramadan effect within Muslim majority countries and regions.

To take an account for Heteroscedasticity in the error term, we used the GARCH model. According to Panel 2, we find that the parameters $\alpha'$ and $\beta'$ of ARCH and GARCH are statistically significant and all most case, that indicate volatility clustering effect. Furthermore the sum of GARCH and ARCH parameter are near of one indicating volatility persistence.

## 4.2. Persistence of Performance

In order, to clarify the phenomenon of persistence of performance of HSBC portfolio, we have applied a measurement model, which inspired by Grinblatt and Titman (1992). This regression is basically used to find out if past performance is a good predictor of future performance. This regression is presented as follows:

$$R_{p(t+1)} = \alpha''_p + \beta''_p R_{p(t)} + \varepsilon_p \tag{5}$$

where $R_{p(t+1)}$ *et* $R_{p(t)}$ represent the performance of each portfolio *p* in the periods $t + 1$ and *t*, respectively. The hypotheses tested were:

H'0: $\beta''_p = 0$

H'1: $\beta''_p \neq 0$

Thus, if we accept the null hypothesis, there is no relationship between performance $(t + 1)$ and performance $(t)$.

Alternatively, if we reject hypothesis H'0, there is a relationship between the performance $(t + 1)$ and the performance $(t)$.

Table 6 shows the performance persistence of 15 HSBC portfolios funds over a period from 2011 to 2018, measured through the simple regression presented previously in Equation (4). By looking at the coefficient $\beta''_p$ we can conclude the performance of next month, as well as the persistence of performance.

**Table 6.** Performance persistence.

| Portfolios | $\alpha''$ | $t(\alpha'')$ | $\beta''$ | $t(\beta'')$ | R2 |
|------------|-----------|---------------|-----------|--------------|-----|
| LC | 0.3165 | 0.4737 | 0.0182 | 0.1678 | 0.0003 |
| LS | 0.0226 | 0.0440 | −0.1887 *** | −4.0294 | 0.0358 |
| AS | −0.0316 | −0.3010 | −0.4484 *** | −15.4669 | 0.2034 |
| IC | 0.0604 ** | 1.7430 | 0.3017 *** | 4.5690 | 0.0832 |
| IS | 0.1964 | 0.4757 | 0.1638 | 1.4494 | 0.0268 |

Note: this table estimates the persistence performance of the 5 portfolios through the coefficients $\beta''$ and $\alpha''$. We used a simple regression: $R_{p(t+1)} = \alpha''_p + \beta''_p R_{p(t)} + \varepsilon_p$. The model is significant at * 10%, ** 5% and *** 1%.

According to the table below, we note that all portfolios managed by HSBC Saudi mutual funds present a coefficient $\beta''_p$ different than zero; for this reason, we are able to reject hypothesis H'0 that

implies that, there is a relationship between the performance ($t + 1$) and the performance ($t$). In spite of that this relationship does not give an answer about the existence of persistence.

From Table 6 we can show that 50% of portfolios have a significant $\beta''_p$ coefficient at the 10 per cent level, as well as: international-focused conventional portfolio (IC), Arab sharia (AS) and Local Sharia (LS). Despite that, the international-focused conventional portfolio (IC) is the unique portfolio that has positive significant coefficients this means that, IC portfolio present a persistent performance during eight years. The managers of these portfolios have a certain degree of predictability of future performance, so they are able to maintain a certain stability in their performance. Moreover, the existence of performance persistence of (IC) confirms that the managers of this portfolio are encouraged to make their investment choices based on the historical performance. The performance persistence finding is considered as a test of manager's market ability to underperform the market index. Finally, our empirical results reported in Table 6 provide empirical evidence to reject hypothesis H3, because only a special case of conventional portfolios (as well as the IC portfolio) are able to present a stable performance and all Islamic portfolios are not able to perform significantly.

## 5. Conclusions

The main objective of this paper is to compare investment strategies and financial performance persistence of Islamic and conventional Saudi funds managed by HSBC Saudi Arabia limited for the period 2011–2018.

The study used a portfolios approach to help diversify away funds specific risks. The Saudi HSBC mutual funds are grouped into equally balanced portfolios and divided into five subgroups based on geographical focused and Sharia compliance: Local/Islamic, local/conventional, global/Islamic and global/conventional and Arabic Islamic.

The preliminary empirical findings of this work show that all international, local and Arab portfolios underperformed their corresponding benchmarks with a smaller mean. We showed that Saudi Islamic mutual funds managed by HSBC underperformed their conventional counterparts. However, to conclude about active management strategy we have using a qualitative method through a model based on Ferson and Schadt (1996) model, with different informational variables as well as a Ramadan effect and treasury bills.

We found that the selectivity skills of manager in the Islamic Saudi HSBC portfolios are stronger and they have better selectivity skills than the conventional Saudi HSBC portfolios. The local Islamic portfolios are highly competitive and generate good returns with reasonable risk. The manager of HSBC Islamic portfolio (LS) is able to take advantage of earnings anomalies, and thus, they make the right choice in the selection of securities. These portfolios are able to generate higher excess returns than conventional Saudi HSBC portfolios.

In terms of market timing ability, the finding from this article showed that local Islamic and conventional HSBC portfolios did not exhibit a market timing ability when using different market benchmarks. However, the international-focused conventional HSBC portfolios possess significant and modest market timing ability; thus, they are able to beat the market index by predicting its movements and buying and selling accordingly. The other international HSBC portfolios have a negative market timing, which can be interpreted as a failure of the fund manager to beat the market and their ability to do the opposite of the market. The international HSBC portfolios focused Sharia increase (decrease) their market exposure in question at a time when the market is down (up).

According to this study, we can note that 20% of HSBC portfolios present a performance persistent over eight years. The managers of these portfolios have a certain degree of predictability of future performance, thus, they are able to maintain a certain stability in their performance.

The choice of benchmark in this paper might not be ideal. Dellva et al. (2001) found quite different results through analysis of the selectivity and market timing estimates with respect to three distinct benchmarks, which can show that the user market index has the ability to incorporate biases in estimates. Hence, in the future work, it will be interesting to use other benchmarks. In summary,

in future empirical works, we can use longer data, a larger sample of Saudi mutual funds and we can also incorporate new variables that allows to understand to what extent Islamic law affects mutual funds active management strategies and the performance.

**Funding:** The author gratefully acknowledge the approval and the support of this research study by the grant from the Deanship of Scientific Research at Northern Border University, Arar, K.S.A.

**Conflicts of Interest:** The author declares no conflict of interest. The views expressed in the paper are attributed entirely to the author and do not necessarily reflect the views of the places at which they work.

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
