# Peer review of "Selectivity and Market Timing Ability of Fund Managers: Comparative Analysis of Islamic and Conventional HSBC Saudi Mutual Funds"

_ijfs, doi:10.3390/ijfs7030048_

Round 1
Reviewer 1 Report
Referee Report for
“Selectivity and Market Timing Ability of fund managers: comparative analysis of Islamic and conventional Saudi mutual funds”
(ijfs-530085)
This paper examines and compares the selectivity and market timing abilities among Islamic and conventional mutual funds in Saudi Arabia. I really like section 2 since the author provides a clear and informative literature review. However, there are some questions about methodology and empirical results in concurrent manuscript. Therefore, in my opinion, this paper is must be improved substantially in order to meet the standard of IJFS. So, I list my suggestions for revising this paper as the follows.
Comments
1. This paper selects 15 mutual funds managed by HSBC Saudi Arabia. In fact, there are 38 fund managers (fund families) operating in Saudi Arabia. Is there any reason why the author only choose HSBC Saudi Arabia? Accordingly, I have two additional questions about sample selection
(1) There is an issue related to mutual-fund families bias. Some literature address several within-family effects, such as intrafirm competition and stellar effect (Nanda et al., 2004; Kempf and Ruenzi, 2008). Hence, I cannot realize why the author only choose HSBC Saudi Arabia.
(2) The total number of mutual fund sample is only 15. The sample may not be representative and the research conclusions cannot be generalized to whole mutual fund industry in Saudi Arabia.
2. In section 4, the author just presents the results. However, it would be better that the author can further discuss and compare these findings with existing literatures.
3. On line 535, the author says that most portfolios have an insignificant β’’ coefficient. Actually, there are two portfolios present significantly negative β’’ coefficients, while the author does not discuss them.
4. There are lots of typos. For examples,
(1) In Equations (1) and (2), the left-hand side variables disappear.
(2) Line 92. “Sock Picking” should be “Stock Picking”.
(3) Line 116. “This funding” should be “This finding”.
(4) Line 160. “risk-adjested” should be “risk-adjusted”.
(5) Lines 186, 277, 280, and so on. Please check whole manuscript again.
Suggested References
1. Nanda, V., Wang, Z. J., & Zheng, L. (2004). Family values and the star phenomenon: Strategies of mutual fund families. The Review of Financial Studies, 17(3), 667-698.
2. Kempf, A., & Ruenzi, S. (2007). Tournaments in mutual-fund families. The Review of Financial Studies, 21(2), 1013-1036.
Author Response
1.This paper selects 15 mutual funds managed by HSBC Saudi Arabia because:
a) HSBC is very experienced financial institution and has a good reputation managing funds in the world.
b)For reasons of lack of information on the period studied for the most Saudi funds, we had to choose a funds how exhibits a complete information during this period.
c) Moreover, HSBC funds are well balanced and they can be divided into 4 sub-groups, namely, Local, Global, Islamic, and Conventional, consisting of the specific purpose of the present
study. The funds managed by HSBC are diversified in terms of investment goal classification, geographical focus and security type, so these funds can summarize us or give us a general view of the Saudi funds operating during 2008-2018.
d) Many previous studies focused on only one fund families as well as Merdad, Hassan, and Alhenawi (2010)., Weber and Ang (2016),Omri et al.(2018).
1. Merdad, Hesham., M. Kabir. Hassan, and Y. Alhenawi. 2010. Islamic versus Conventional Mutual Funds Performance in Saudi Arabia: A Case Study. Journal of King Abdulaziz
2. Omri, Abdelwahed., Sassou, Karim. And Ben Sedrine Goucha Nadia. 2018.On the post-financial crisis performance of Islamic mutual funds: the case of Riyad funds. applied economics 51: 1929-1946
e) The objective of our study is to following and to expanding the work of Merdad, Hassan, and Alhenawi (2010). The previous study of Merdad, Hassan, and Alhenawi (2010) investigates the impact of Shari’ah screening on the performance of HSBC mutual funds. However our paper expand their results by examining the selectivity and market timing abilities among Islamic and conventional HSBC funds.
(I will add a paragraph that explains in detail this choice).
conclusions cannot be generalized to whole mutual fund industry in Saudi Arabia :
That’s true. I will mention in the conclusion that the results imply only a specific category of Saudi funds.
2. In section 4, the author just presents the results. However, it would be better that the author can further discuss and compare these findings with existing literature:
I will reformulat the entire results and I will add a discussion of the emperical finding.
3. There are two portfolios present significantly negative β’’ coefficients: we will discuss them.
4. There are lots of typos : I will correct them.
Reviewer 2 Report
The paper is on time skills and performance of mutual funds in case of Islamic versus conventional funds. The subject is interesting but the paper needs improvement based on my comments below.
The authors should clearly present the main motivation of the paper in the introduction and refer to the main contribution of their findings.
Literature review can be enriched by including other relevant papers
· Christensen, M. (2005). Danish mutual fund performance-selectivity, market timing and persistence. Aarhus School of Business, Finance Research Group Working Paper No. F-2005-1.
· Papadamou, S., & Siriopoulos, C. (2004). American equity mutual funds in European markets: hot hands phenomenon and style analysis. International Journal of Finance & Economics, 9(2), 85-97.
· Papadamou, S., Kyriazis, N., & Mermigka, L. (2017). Japanese mutual funds before and after the crisis outburst: A style-and performance-analysis. International Journal of Financial Studies, 5(1), 9.
· Tan, Ö. F. (2015a). Mutual Fund Performance: Evidence from South Africa. Emerging Markets Journal, 5(2), 49.
· Tan, Ö. F. (2015b). The Performance of Indian Equity Funds in the Era of Quantitative Easing. International Journal of Commerce and Finance, 1(1), 11-24.
In Table 1 AS seems to be double-counted.
In table 5 the authors put asterisks on coefficients with high positive t-stats but not on negative ones that are also statistically significant
In equation 1 and 2 R is located in wrong order please correct it.
In equation 4 the author refers to ???and ??? but in equation subscript p is missing.
The author argues that the ???−?? Is the return of 3 month treasury bills for the previous years. However, from this measure substract return of the market in equation 1 and 4 without explaining what this excess return shows.
I think that is better to see any persistence effect in model in equation 4 by including a lagged dependent variable. Because there are serious omitted variable problems otherwise (see equation 5 again).
Correct some typos and English in some cases like
· “Our 20 results showed that Islamic funds that apply Sharia law can not generate constant 21 performance and management capacity over time. while Saudi conventional funds focused”
· 22 internationally have a persistent performance over the long term.page 15 “the futur empirical researchs”
English needs improvements.
Author Response
1- The authors should clearly present the main motivation of the paper in the introduction and refer to the main contribution of their findings.
I will add a motivation in introduction
“The most empirical research that investigated about Islamic mutual funds still do not provide a definite answer to this most critical question: are the management ability of manager in Islamic mutual funds different from the conventional fund? Globally, this insufficiency motivated us to compare the performance persistence and management capacities of Islamic mutual fund and conventional funds in Saudi Arabia. We are motivated to study Islamic funds in Saudi Arabia for several reasons. First, Saudi Arabia remains the key home countries contributing to the biggest market share of the largest Islamic funds industry. Moreover, Saudi Islamic and other non-Islamic mutual funds have grown considerably. We are going to make a comparative study between this two funds category. Given the lack of information on the period studied for most data from Saudi funds, we had to choose a fund how present complete information spanning from 2011 to 2018. For this reason, in the present study we select the funds managed by HSBC Saudi Arabia. HSBC is very experienced financial institution and has a good reputation managing funds in the world. Moreover, HSBC funds are well balanced and they can be divided into many sub-groups, namely, Local, International, Islamic, and Conventional, consisting of the specific aim of the our study. The funds managed by HSBC are diversified in terms of investment goal classification, geographical focus and security type, so these different category funds give us a general view of some Saudi funds operating during 2008-2018.”
2- Literature review can be enriched by including other relevant papers
I will add these references to my literature Revue.
· Christensen, M. (2005). Papadamou, S., & Siriopoulos, C. (2004).
· Papadamou, S., Kyriazis, N., & Mermigka, L. (2017). · Tan, Ö. F. (2015a).
3- In Table 1 AS seems to be double-counted.
I corrected it.
4- In table 5 the authors put asterisks on coefficients with high positive t-stats but not on negative ones that are also statistically significant
I corrected it, and I changed the interpretation.
5- In equation 1 and 2 R is located in wrong order please correct it.
I corrected it.
6- In equation 4 the author refers to ???and ??? but in equation subscript p is missing.
I changed it.
7- The author argues that the ???−?? Is the return of 3 month treasury bills for the previous years. However, from this measure substract return of the market in equation 1 and 4 without explaining what this excess return shows.
I explained it.
10- Correct some typos and English
I improved English for all paper
Round 2
Reviewer 1 Report
In general, this manuscript has been revised based on what I suggest. However, I still concern about the sample collection issue. The author indicates that some literature focused on only one fund families, such as Merdad et al. (2010) and Omri et al.(2018). As we can see the titles of these papers, they used the term - "a case study" or "the case of Riyad funds", because they only focused on one fund family. Therefore, I strongly suggest that the author can modify the title of this paper to identify that this paper only applies the data for HSBC Saudi Arabia.
One more minor concern is that, on line 644, the author mention "This finding provides empirical evidence that these 3 portfolios present a performance persistent during 8 years". In my opinion, negative coefficients do not mean performance persistent. The author should revise this discussion.
Author Response
I change the title :
Selectivity and Market Timing Ability of fund managers: comparative analysis of Islamic and conventional HSBC Saudi mutual funds
2. Negative coefficients do not mean performance persistent. The author should revise this discussion: I change the discussion.